# Extracellular Vesicle-Associated miRNAs and Chemoresistance: A Systematic Review

**DOI:** 10.3390/cancers13184608

**Published:** 2021-09-14

**Authors:** America Campos, Shayna Sharma, Andreas Obermair, Carlos Salomon

**Affiliations:** 1Exosome Biology Laboratory, Center for Clinical Diagnostics, University of Queensland Centre for Clinical Research, Royal Brisbane and Women’s Hospital, The University of Queensland, Brisbane, QLD 4029, Australia; america.camposg@gmail.com (A.C.); shayna.sharma@uq.net.au (S.S.); 2Queensland Centre for Gynaecological Cancer Research, The University of Queensland, Centre for Clinical Research, Building 71/918, Royal Brisbane and Women’s Hospital, Herston, QLD 4029, Australia; a.obermair@uq.edu.au; 3Departamento de Investigación, Postgrado y Educación Contínua (DIPEC), Facultad de Ciencias de la Salud, Universidad del Alba, Santiago 8370007, Chile

**Keywords:** exosomes, miRNAs, cancer

## Abstract

**Simple Summary:**

There is an urgent need for a non-invasive, specific biomarker to identify patients at risk of chemoresistance, which it is the ability of cancer cells to escape the effect of chemotherapy drugs. Extracellular vesicles contain an abundance of miRNAs that demonstrate expression across a range of cancers including breast cancer, renal cell carcinoma, lung cancer, multiple myeloma, and lymphoma. Interestingly, miRNAs encapsulated within extracellular vesicles (EVs) including exosomes display an association with chemoresistance. Here, we performed a systematic revision to evaluate the association between miRNAs within EVs and chemotherapy resistance. The summarized graphical abstract indicates that several exosome-derived miRNAs involved in chemotherapy resistance can be found among different types of cancers, such as colorectal, ovarian, breast, and lung cancer, and lymphoma.

**Abstract:**

Cancer is a leading public health issue globally, and diagnosis is often associated with poor outcomes and reduced patient survival. One of the major contributors to the fatality resultant of cancer is the development of resistance to chemotherapy, known as chemoresistance. Furthermore, there are limitations in our ability to identify patients that will respond to therapy, versus patients that will develop relapse, and display limited or no response to treatment. This often leads to patients being subjected to multiple futile treatment cycles, and results in a reduction in their quality of life. Therefore, there is an urgent clinical need to develop tools to identify patients at risk of chemoresistance, and recent literature has suggested that small extracellular vesicles, known as exosomes, may be a vital source of information. Extracellular vesicles (EV) are membrane bound vesicles, involved in cell-cell communication, through the transfer of their cargo, which includes proteins, lipids, and miRNAs. A defined exploration strategy was performed in this systematic review in order to provide a compilation of key EV miRNAs which may be predictive of chemoresistance. We searched the PubMed, Science Direct, and Scopus databases using the following keywords: Extracellular vesicles OR exosomes OR EVs AND miRNA AND Chemotherapy OR Chemoresistance OR Cancer Recurrence from 2010 to 2020. We found 31 articles that reported key EV-associated miRNAs involved in cancer recurrence related to chemoresistance. Interestingly, multiple studies of the same tumor type identified different microRNAs, and few studies identified the same ones. Specifically, miR-21, miR-222, and miR-155 displayed roles in response to chemotherapy, and were found to be common in colorectal cancer, ovarian cancer, breast cancer, and diffuse large B cell lymphoma patients (DLBCL). miR-21 and miR-222 were found to favour the development of chemoresistance, whereas miR-155 exhibited a contrasting role, depending on the type of primary tumor. Whilst high levels of miR-155 were found to correlate with chemotherapy resistance in DLBCL, it was found to be predictive of an effective response towards chemotherapy in breast cancer. Thus, further research regarding the roles of these miRNAs would be beneficial in terms of designing novel tools to counteract the progression of cancer in a not-to-distant future.

## 1. Introduction

Chemotherapy and targeted drug resistance are major barriers in cancer therapy, with several mechanisms leading to drug resistance, including increased drug efflux, drug target mutations, and interference in regular cellular processes such as apoptosis [1]. Chemoresistance has previously been associated with genetics; however, recent literature has demonstrated that it is in fact due to both genetic and epigenetic factors. Studies have found that genetics alone is insufficient to explain the rapid occurrence of resistance to treatments. Additionally, resistance has been found to be reversible, and it is highly likely that epigenetic factors are involved in facilitating resistance [2]. With the rise of interest in treatment resistance, the question of whether chemotherpy resistance develops prior to treatment, or upon exposure to treatment, has come to the forefront. Thus, there is an urgent need for identifying non-invasive and reliable biomarkers that can predict a patient’s response to specific chemotherapeutics. 

miRNAs are a highly conserved, small non-coding class of RNAs, of approximately 25 nucleotides in length [3]. They are capable of binding to complementary sequences in the 3′untranslated region of their target mRNAs, thus leading to mRNA inhibition [4]. Moreover, these RNA species can also be released by active secretion into the blood stream in response to several physiological processes, including cell death events, such as apoptosis or necrosis [5]. Furthermore, specific miRNAs have been described as having differential expression in tumor tissues [6]. Interestingly, miRNAs also display remarkable stability in biofluids. They can travel throughout the blood stream as “free” miRNAs associated with protein complexes (for example, argonaute 2 protein) [7], or as an encapsulated form within different subtypes of extracellular vesicles (EVs), such as exosomes, microvesicles, or apoptotic bodies, which can protect these miRNAs from RNAse activity, thus preventing degradation [5,8,9].

Extracellular vesicles (EVs) are a heterogeneous population of membrane-enclosed, non-replicating, and sub-micron sized structures, which are actively secreted by a wide variety of eukaryotic and prokaryotic organisms [10,11]. EVs are classified into exosomes, microvesicles, or apoptotic bodies, depending on their molecular content which may include proteins derived from their biogenesis pathways. In the context of exosomes, these are mainly identified by the presence of tetraspanins, such as CD81 and CD9, or proteins associated with the key mediators of multivesicular body biogenesis, known as the endosomal sorting complex required for transport (ESCRT) pathway, which includes proteins such as TSG101 or Alix. In terms of size, EVs can range between 40 to >1000 nm in diameter and can be quantified and visualized by Nanoparticle Tracking Analysis and Transmission Electron Microscopy, respectively [11]. In addition, EVs can function as mediators of communication between cells in physiological and pathological settings and are also capable of carrying a diverse array of biomolecules, such as lipids, nucleic acids (e.g., miRNAs), carbohydrates, and proteins [11,12]. Among the many currently known EV subtypes, exosomes have garnered significant attention in the context of chemotherapy resistance, since several reports have demonstrated that they are capable of conferring chemoresistance to recipient cells through the transfer of miRNAs, as presented in Figure 1 [1,13,14]. Conversely, there are several exosome-derived miRNAs which are found to be downregulated in biological fluids, such as serum, and this down-regulation is correlated with the development of chemotherapy resistance in specific types of cancers [15,16,17]. 

Hence, the aim of this systematic review is to corroborate the importance of circulating miRNAs associated with extracellular vesicles (EVs) or exosomes as potential predictors of chemotherapy resistance, by examining the available literature. 

## 2. Materials and Methods

### 2.1. Search Strategy

A comprehensive systematic literature search was performed to identify studies up to 9 September 2020, which assessed the role of exosome-derived miRNAs involved in predicting chemoresistance in advanced cancer patients. In addition, this review was performed following the guidelines outlined in the Cochrane Handbook for Systematic Reviews of Interventions [20]. We used three search engines: PubMed, ScienceDirect, and Scopus. Chosen reporting items for the Systematic Review statement were used as the reference standard [21]. The search strategy was as follows: Extracellular vesicles orexosomes or EVs and miRNA and Chemotherapy or Chemoresistance or Cancer Recurrence. 

### 2.2. Eligibility Criteria

We only considered articles that were written in English and excluded duplicates or articles that were found in common among the three selected search engines. Next, we excluded non-original articles such as reviews, encyclopedia-derived articles, book chapters, conference abstracts, discussion articles, editorials, mini-reviews, and meta-analyses (Figure 2). Subsequently, titles and abstracts of all relevant studies were evaluated carefully and screened, before downloading the full-text article; we screened for studies which included miRNAs, exosomes, or extracellular vesicles, as well as information on chemotherapy resistance as the main focus of research. In addition, studies presenting clinical data on patients treated with chemotherapy, or in vitro preclinical analyses that were later validated on treated or relapsed cancer patients, were also considered. Information regarding cancer type, EV-derived miRNAs, chemotherapeutic drug, biofluid, exosome or EV isolation method, and the miRNA platform that was used, were all taken into consideration. Moreover, studies were removed based on the following exclusion criteria, which included studies with only in vitro experiments, and/or animal model-based results, studies focusing on other non-coding RNA besides miRNA, studies focusing on radiotherapy, and studies focusing on prokaryotic organisms.

### 2.3. Data Extraction

Two investigators (AC and SS) independently filtered the relevant articles which satisfied the eligibility criteria. Information regarding first author, publication year, exosomal miRNAs, chemotherapeutic drug, patient information (including cancer type, country of origin, gender, biofluid, and sample size versus control (healthy subjects, benign disease or drug-sensitive cancer patients)), and type of platforms that were used, including EV or exosome isolation method, and miRNA profiling platform information and statistical data (*p*-value or ROC for AUC for example), were all retrieved by the two investigators independently. Results indicating a *p*-value greater than 0.05 were not considered. Any inconsistency was resolved by revisiting the data and discussion among the authors. 

## 3. Results

### 3.1. Literature Search and Characteristics

The initial literature search yielded a total of 442 articles in PubMed, 1428 articles in ScienceDirect, and 1967 articles in the Scopus database. After carefully screening the titles and abstracts of each article, in which reviews, mini-reviews, encyclopedia-derived articles, book chapters, conference abstracts, discussion articles, editorials and meta-analyses were not further considered, a total of 409 articles from PubMed, 1420 from ScienceDirect, and 1961 from the Scopus database search were excluded; 47 articles were used for subsequent analysis and were obtained as full-text articles. Further screening rendered 37 relevant articles, since 9 articles were found to be common amongst the search engines, and one article only on radiotherapy-associated results. Additional in-depth revision of the selected articles led to the exclusion of six more articles which did not focus on the role of exosomes in the development of chemoresistance in advanced cancer patients (Figure 2 and Table 1).

Thirty-one articles were gathered in the final selection process (Figure 2) and were mainly based on retrospective, cohort, and case-control studies (Table 1). In addition, these articles presented pre-clinical and clinical data that was later validated on chemotherapy-treated patients with either renal [22], lung [1,15,23,24], colorectal [25,26,27], glioblastoma [28,29], prostate [14,30], hepatocellular carcinoma [17,31], ovarian [4,32,33,34,35,36], breast [37,38,39,40], multiple myeloma [41], lymphoma [42,43,44], gastric [45], pancreatic cancer [46], or melanoma [47]. Interestingly, four of these selected articles (Figure 2 and Table 2) suggest that EV- or exo-miRNAs may predict the emergence of chemotherapy resistance [1,15,22,23], and that they may have a role in transferring drug resistance, via shuttling miRNAs. However, additional data (in vivo experiments using these EVs or exosomes, for example) and a larger and variable cohort of patients that respond effectively to chemotherapy, is needed to confirm whether these candidate exosomal miRNAs do indeed play a role in regulating drug sensitivity. 

Nonetheless, studies have evaluated multiple miRNA candidates (Table 1) [25,28] obtained from tumor cell lines (and exosomes derived from these cell lines) already exposed to different chemotherapeutic drugs, and compared them to their parental tumor cell line (and exosome) counterpart [30,31,32]. Once these candidates were selected, authors assessed their clinical relevance in tumor tissues or serum samples, from patients [17,28] with cancer (Table 2). The association between exosomal miRNAs and chemoresistance in multiple different types of cancers has been established in the literature, and here we report the key findings from these studies, according to tumor type.

### 3.2. Candidate Exosomal-Associated miRNAs Involved in Chemotherapy Response 

#### 3.2.1. Colorectal Cancer

Three research articles were found amongst the prospective studies that evaluated the importance of several candidate miRNAs in advanced colorectal (CRC) cancer patients treated with chemotherapeutic drugs such as Oxaliplatin, Leucovorin, 5-fluorouracil, and/or Bevacizumab [25,26,27] (Table 3). Only the studies of De Miguel Perez and Jin evaluated the role of EV-derived miRNAs obtained directly from the serum samples of a cohort of patients which ranged up to 44 individuals. Contrastingly, Liu and collaborators evaluated the correlation of a specific miRNA in tumor biopsies of CRC patients. In addition, their focus was on the in vitro and in vivo role of their target miRNA present in EVs derived from CRC cell lines which could enhance chemosensitivity in resistant cancer cells [25]. This highlighted that fact that differences may arise when evaluating miRNAs isolated from serum-derived EVs compared to miRNAs directly obtained from tumor tissue.

de Miguel Perez and collaborators compared a specific panel of EV-isolated miRNAs (miR-19b, miR-21, miR-222, and miR-92a) from 44 metastatic CRC cancer patients treated with standard first–line chemotherapeutic treatment (FOLFOX-6m), and an antiangiogenic therapy (Bevacizumab), versus 17 healthy subjects. They found that this specific set of miRNAs were elevated in exosomes obtained from patients, however there was no association with treatment response [26], thus indicating that this specific set of EV-derived miRNAs may display an important potential in terms of CRC prognosis in these patients.

Alternatively, Jin and collaborators reported a set of miRNAs identified from an in vitro study involving the comparison of oxaliplatin/5-fluorouracil-resistant CRC cell lines and the corresponding exosomes, against their parental CRC cell lines and exosomal counterparts [27]. Interestingly, the authors found that miR-21-5p, miR-96-5p, miR-1246, and miR-1229-5p were found to be present in the exosomes derived from the serum of chemoresistant patients (*n* = 25) when compared to the chemosensitive group (*n* = 18), as well as in the resistant cell lines. Moreover, a synergistic effect of the four miRNAs was observed (AUC = 0,804, *p* < 0.01) when compared to the effects observed from a single miRNA candidate. These results suggested that a select panel of chemotherapy resistance markers may improve the power of cancer prognosis when compared to the use of a single biomarker [27,49]. This is one of the first studies which identifies a specific set of exo-miRNAs that may distinguish the chemoresistant group from late-stage CRC patients; however, a larger cohort (>100 patients) of advanced CRC patients’ is required. 

As mentioned above, the reasons why the study by Liu and collaborators was included in this review relied on the fact that both their in vitro and in vivo studies demonstrated that miR-128-3p was found to be downregulated in oxaliplatin-resistant CRC cell lines and, even more importantly, authors demonstrated that the exosome-mediated transfer of miR-128-3p can promote Oxaliplatin sensitivity in resistant cells and could also restore Oxaliplatin drug response via intra-tumor injection in vivo. In addition, Liu and collaborators complemented their research by analyzing the presence of miR-128-3p in tumor biopsies obtained from several cohorts of patients with advanced colorectal cancer (CRC). The study included a training phase (*n* = 51) and a validation phase (*n* = 122), and samples were distinguished according to the absence or presence of a specific chemotherapeutic treatment (Oxaliplatin), or development of cancer relapse during treatment [25]. Altogether, the authors suggested an inverse correlation between miR-128-3p expression and progression-free survival of advanced CRC patients treated with Oxaliplatin. Most importantly, the delivery of miR-128-3p via exosomes has the potential to be used as a tool to counteract chemoresistance in these patients. 

#### 3.2.2. Ovarian Cancer

Six studies evaluated the role or presence of miRNAs involved in chemoresistance or chemosensitivity response in ovarian cancer patients by evaluating the role of these miRNAs in EVs found in vitro [5,32,33,34,35,36] (Table 3). Alharbi and colleagues sought to evaluate the presence of miR-891-5p, which was one of their selected EV-associated miRNAs found to be positively associated with aggressive and chemoresistant ovarian cancer cell behavior in response to Carboplatin in vitro. Interestingly, SWATH mass spectrometry analysis gave insights into the role of miR-891-5p, whose overexpression correlated with elevated expression of DNA repair-associated proteins and MYC targets [33]. Although the role of this miRNA was not evaluated in vivo, here the authors efficiently screened the presence of miR-891-5p in small EVs found in the plasma of 17 ovarian cancer patients before surgery. After follow-up, results of this prospective study found that patients with recurrent disease (*n* = 6) displayed elevated levels of EV-associated miR-891-5p compared to patients which were alive without disease (*n* = 11), therefore suggesting that the presence of this EV-associated miRNA may potentially aid in monitoring ovarian cancer patient response towards chemotherapy [33]. 

Kanlikilicer and collaborators on their behalf explored the role of exosomal miR-1246, since a retrospective study of tissue samples (*n* = 138) obtained from the Tumor tissue and Cancer Genome Atlas (TCGA) found that high expression of this miRNA was significantly associated with worse overall prognosis compared to patients which displayed low but detectable levels of miR-1246. Specifically, the authors demonstrated that this miRNA is highly expressed in ovarian cancer exosomes compared to their cells of origin, and the use of an anti-miR-1246 treatment along with the overexpression of Caveolin-1 (a miR-1246 target) was capable of sensitizing ovarian cancer cells to Paclitaxel. Most importantly, the authors revealed that the combined use of a miR-1246 inhibitor and Paclitaxel in an intraperitoneal orthotopic ovarian cancer model led to reduced tumor burden in vivo, despite the fact that Kanlikilicer and colleagues did not isolate exosomes from patient samples but only evaluated the presence of miR-1246 and exosome markers such as CD63 in ovarian tumor samples (*n* = 15), as well as in normal ovarian surface epithelium samples (*n* = 7). Results demonstrated that miR-1246 strongly colocalized with CD63 in tumor samples [32]. Altogether, this data suggests that the inhibition of circulating miR-1246 along with Paclitaxel treatment may serve as an approach to chemosensitize and treat patients with resistant ovarian cancer.

The drug-resistant role of exosomal miRNAs in plasma or serum samples obtained from ovarian cancer patients, treated with platinum-based therapies, has also been established. Interestingly, Kuhlmann and collaborators acknowledged the importance of identifying novel EV-associated miRNAs which may be influencing the development of chemoresistance or chemosensitivity in these patients, and, therefore, focused on designing a robust and easy-to-use workflow to analyze these miRNAs by Next Generation Sequencing (NGS) [5]. To this end, the authors selected participants from a retrospective study that included fifteen platinum-resistant ovarian cancer patients that had recurred within six months after completing adjuvant platinum-based chemotherapy, and compared them against fifteen platinum-sensitive ovarian cancer patients. Subsequently, the authors isolated EVs from the plasma of these patients and determined that the use of the ExoQuick reagent efficiently gave an optimal EV RNA yield and a reproducible small RNA library, when compared to the use of ExoRNeasy as an EV isolating kit. Consequently, the authors identified a distinct profile of differentially expressed mature miRNA sequences (miR-181a, miR-1908, miR-21, miR-486, and miR-223) in EV samples from the group of platinum-resistant ovarian cancer patients. However, these results were not statistically significant, since this selected set of EV-associated miRNAs did not share any correlation with the patient’s clinicopathological data. Authors discussed that this result may be due to the use of a limited number of samples and due to the complexity of the plasma sample itself [5]. Nonetheless, this distinct set of EV-associated miRNAs has been previously correlated with platinum resistance and ovarian cancer [50,51,52,53], as seen in other studies mentioned in depth in this review [35,36]. 

Exosomal miR-223 is an example of the former, since this non-coding RNA has been identified as being abundant in plasma and serum samples obtained from platinum-resistant or recurrent ovarian cancer patients [5,36] compared to platinum-sensitive patients [5] or matched ovarian cancer patients who had recently undergone surgery [36]. Initially, Zhu and collaborators evaluated the presence of miR-223 in 62 ovarian tumor tissue samples and revealed that 33 samples that were obtained from cisplatin-resistant patients displayed higher levels of miR-223 and displayed a lower expression of PTEN (phosphatase and tensin homologue), which is a known tumor suppressor gene and one of the identified targets of miR-223 [36,54]. Altogether, these results led the authors to establish that a longer progression-free survival of ovarian cancer patients was significantly correlated with a higher expression of PTEN and a lower expression of miR-223. Most importantly, Zhu and collaborators also focused on evaluating the presence of miR-223 in exosomes from a retrospective study, which included the sera of ovarian cancer patients (*n* = 12) who had been treated with cisplatin and taxol after their first surgery and compared them against their matched patient samples (*n* = 12) that had relapsed within or after 6 months of their last cycle of chemotherapy. Results demonstrated that higher levels of exosomal miR-223 were widely detected in recurrent or relapsed patients when compared with their paired samples before relapse. These results were strengthened through gain and loss-of-function experiments, since Zhu and colleagues also revealed that exosomal miR-223 could promote drug resistance in epithelial ovarian cancer cells via regulation of PTEN-PI3K/AKT pathways, both in vitro and in vivo. The data thus supports the fact that miR-223 could be used as a predictive marker of drug resistance and may be considered a possible drug target to overcome chemoresistance in ovarian cancer patients. 

Furthermore, miR-21 [5] was also identified as being expressed in exosomes obtained from primary cultures of adipocytes and fibroblasts derived from ovarian cancer patients, and patients with benign gynecological diseases (control group) [35]. It was found that exosomal miR-21 can be transferred from stromal cells to ovarian cancer cells in vitro and may confer a more aggressive and chemoresistant phenotype in vitro and in vivo (a subcutaneous model of ovarian cancer treated with Paclitaxel). In addition, the authors mentioned that this chemoresistant phenotype may be promoted by the downregulation of apoptotic protease activating factor 1 (APAF1), which has been recently described as a novel target of miR-21. These results, therefore, imply that focusing on the upregulation of APAF in ovarian cancer cells may be used as a possible way of sensitizing ovarian cancer cells to paclitaxel treatment [35].

While Pink and colleagues initially focused on performing microarray analyses to identify which miRNAs were capable of inducing resistance to cisplatin in vitro, their results demonstrated that the passenger strand of miR-21 (miR-21-3p) of the cisplatin-resistant ovarian cancer cell line CP70 was responsible for increasing cisplatin resistance in several ovarian cancer cell lines. Additionally, the authors isolated and evaluated the biological effect of exosomes derived from CP70 cells and revealed that these EVs could also enhance cisplatin resistance in recipient ovarian cancer cell lines. Nonetheless, further studies are required since the authors were not capable of attributing this effect to the sole presence of miR-21-3p in EVs when being delivered to recipient cells [34]. Interestingly, Pink and colleagues also explored transcriptomic data from a previously published work [55], that was already available in the Gene Expression Omnibus (GEO) repository database (42 ovarian tumor tissue samples) and demonstrated that miR-21-3p increased in chemoresistant ovarian tumors (compared to chemosensitive tumors). In this regard, authors also found retrospective data from another publicly available transcriptomic database, which is TCGA, and demonstrated that this miRNA was associated with a shorter platinum free interval (PFI) when compared to tumors with low expression of miR-21-3p (282 tumor samples), thus complementing the aforementioned results that suggest that an increased expression of this miRNA might be correlated with more resistant ovarian tumors. 

It is of worth mentioning that data mining from the TCGA database and the GEO repository databases have also been known to be useful tools when establishing the association between other EV-associated miRNAs and platinum-based therapies [30,31,32,55,56]. In fact, these databases contain valuable transcriptomic information from several types of tumor samples that are not limited to ovarian cancer samples, which may include data regarding PFI and information on cancer patient relapse (drug-resistant or drug-sensitive tumors). 

#### 3.2.3. Breast Cancer

Four studies were included in this section (Table 3), which focused on studying the presence and possible chemoresistant or chemosensitive role of circulating or exosome-derived miRNAs in breast cancer patients undergoing neoadjuvant chemotherapy [37,38,39,40]. 

Stevic and colleagues analyzed a real-time PCR-based microRNA array card containing 384 different miRNAs in exosomes obtained from the plasma of a cohort of breast cancer patients (*n* = 15), treated with or without chemotherapy, and with or without pathological complete response (pCR) [37]. pCR refers to the absence of invasive cancer in breast tissue after finalizing neoadjuvant therapy, and therefore this clinical term positively correlates with both increased overall survival and disease-free survival of breast cancer patients [57]. 45 miRNAs were found to be significantly deregulated among the exosomal samples. Subsequently, the expression of these exosomal miRNAs was determined in a much larger cohort of breast cancer patients (*n* = 435) (plasma samples obtained prior to neoadjuvant chemotherapy) and compared to a cohort of healthy subjects (*n* = 20), and/or breast cancer patients (*n* = 9) who had already undergone neoadjuyant chemotherapy before surgery. miR-27a, miR-155, miR-376a, and miR-376c decreased in patients that had undergone treatment compared to samples obtained prior to treatment. Levels of these miRNAs in patients that had undergone treatment were comparable to those observed in healthy controls. The authors suggested that these miRNAs may be released from the primary tumor into the bloodstream, which could reflect, to some extent, the status of the disease in patients that already went through therapy. Furthermore, authors also demonstrated that in uni- and multivariate models, miR-155 (*p* = 0.002 and *p* = 0.003, respectively) and miR-301 (*p* = 0.002 and *p* = 0.001, respectively) significantly predicted pCR in the majority of breast cancer patients that were tested, therefore suggesting an improved response of these patients towards a carboplatin-based therapy. However, the authors stated that it was difficult to make a statistical conclusion regarding the impact of therapy on the expression levels of this selected set of miRNAs, due to the small cohort of patients that had undergone neoadjuvant chemotherapy [37].

An indirect role associated with chemotherapy response was assigned to a group of exosomal miRNAs (miR-21, miR-105, miR-221, and/or miR-222) isolated from the sera of breast cancer patients (*n* = 47) taken before and while undergoing neoadjuvant chemotherapy [38]. Rodriguez-Martínez not only described the presence of exosomal miR-21 as being directly correlated with tumor size (*p* = 0.039), but also revealed that lower levels of this exosomal miRNA were found in breast cancer patients (37 out of 47 breast cancer patients) who were undergoing neoadjuvant chemotherapy compared to untreated patients that displayed exosomal miR-222 along with proliferation markers, such as ki67 (*p* = 0.05). Authors also noted that lower levels of exosomal miR-221 were observed in patients with compromised lymph nodes which were undergoing neoadjuvant chemotherapy (*p* = 0.006) compared to patients with unaffected lymph nodes undergoing the same treatment. As for exosomal miR-105, authors only found that high levels of this exosomal miRNA were found along with exosomal miR-21 in a group of six patients with metastatic disease compared with 8 healthy subjects. Although authors discussed that their focus was to evaluate the role of these exosomal miRNAs as a complementary clinical tool useful for improving breast cancer diagnosis and prognosis, authors also agreed on the fact that the detection of exosomal miR-21, miR-105, miR-221, and/or miR-222 may aid in predicting relapses in the early stages of this disease and may also be used as biomarkers capable of detecting early resistance to doxorubicin/cyclophosphamide treatment [38].

Zhong and colleagues identified a set of 22 miRNAs, which were observed to be upregulated in exosomes derived from breast cancer cell lines resistant to docetaxel, epirubicin, or vinorelbine, compared to exosomes derived from their parental cell lines. Subsequently, the authors evaluated the presence of this selected set of miRNAs in preneoadjuvant chemotherapy biopsies and paired surgically resected specimens of breast cancer patients (*n* = 23), and demonstrated that 12 out of the 22 selected miRNAs were significantly upregulated after pre-neoadjuvant chemotherapy. Most importantly, only miR-574-3p was found to be statistically upregulated in tissues obtained from breast cancer patients with stable and progressive disease compared to tissues obtained from breast cancer patients, which displayed a partial response towards chemotherapy (*p* = 0.027) [39]. 

Interesting in vitro results were found by Bovy and colleagues which revealed that exosomal miR-503 derived from endothelial cells may be capable of displaying a tumor-suppressing role in breast cancer, since their results demonstrated that this exosomal miRNA could decrease the proliferation and invasive potential of recipient MDA-MB-231 breast cancer cells [40]. Moreover, increased levels of miR-503 were found in exosomes isolated from endothelial cells treated with epirubicin and paclitaxel compared to untreated endothelial cells. Similar results were found in a prospective study of breast cancer patients which had been receiving neoadjuvant chemotherapy (*n* = 17), which revealed elevated levels of circulating miR-503 compared to patients who had only undergone surgery (*n* = 12). Although Bovy and collaborators did not isolate exosomes directly from the plasma of breast cancer patients in order to evaluate the role of this miRNA, these results may give insight into what would be happening in the tumor microenvironment in response to chemotherapy, since there are recent studies indicating, for example, that miR-503 can directly bind and inhibit BLACAT1 (bladder cancer-associated transcript 1), which is a long-non-coding RNA that has been found to promote chemotherapy resistance in T47D and MCF7 breast cancer cells [58]. In addition, miR-503 has also been described as part of a miRNA cluster, known as miR-424(322)/503, whose loss in breast cancer patients has been associated with the development of aggressive breast cancer, and also found to promote paclitaxel resistance in vivo due to the increased expression of two of their targets: BCL-2 (B-cell lymphoma 2) and IGF1R (insulin-like growth factor-1 receptor) [59].

#### 3.2.4. Renal Cell Carcinoma

He and colleagues initially reported that EVs derived from sorafenib-resistant renal carcinoma cell lines were capable of transferring drug resistance features to sorafenib-sensitive renal carcinoma parental cell lines [22]. They also reported that among 17 EV-derived miRNAs that were found to be upregulated in resistant-cell lines compared to their parental cell counterpart; only exosomal miR-31-5p was found to display a relevant role in promoting sorafenib resistance according to preliminary data. However, the authors only demonstrated that miR-31-5p was capable of inducing sorafenib resistance in vitro and in vivo by silencing or overexpressing this miRNA in resistant or sensitive parental cell lines, respectively. Interesting data was revealed by the authors in which they could identify MutL homolog 1 (MLH1) as a direct target of miR-31-5p. In addition, the authors demonstrated that the upregulation of MLH1 via lentiviral transduction in resistant cell lines gives rise to a restored sensitivity towards sorafenib in these cell lines. Most importantly, He and colleagues reported a retrospective study in which they displayed the expression of miR-31-5p in EVs derived from the plasma of metastatic renal cell carcinoma (RCC) patients [22]. They compared EVs derived from the plasma of patients (*n* = 40) who had undergone treatment with Sorafenib with EVs from the same patients prior to treatment, and demonstrated that miR-31-5p in EVs is upregulated in patients with progressive disease during sorafenib therapy, when compared to their pre-therapy status. Therefore, these results suggest that the upregulation of this exosomal miRNA in plasma may predict an unfavorable response towards sorafenib therapy in patients with renal cell carcinoma.

#### 3.2.5. Lung Cancer

Four studies were selected in this section (Table 3) that focused on exploring the role of exosomal miRNAs derived from the plasma or sera of patients with non-small cell lung cancer (NSCLC). The upregulation of these selected EV-associated miRNAs in NSCLC patients was significantly associated with poor responses towards gemcitabine [43] and/or platinum-based chemotherapies [1,15,23] compared to NSCLC patients which displayed diminished levels of these candidate EV-associated miRNAs. 

For instance, Wei and collaborators initially revealed that exosomal miR-222-3p was found to be significantly abundant in NSCLC cell lines that were resistant to gemcitabine, compared to their sensitive parental cell line counterpart [43]. Moreover, authors also demonstrated that increased levels of this miRNA were found in recipient sensitive NSCLC lines after being exposed to EVs derived from resistant NSCLC cells. Notably, these recipient cells also displayed an increased resistance to gemcitabine in a dose-dependent manner, which was also accompanied by increased migration and invasion capabilities. These effects were all abolished once miR-222-3p was knocked-down in recipient cells that were afterwards exposed to EVs derived from resistant NSCLC cells. Additional studies also indicated that miR-222-3p targeted and dramatically suppressed the expression of a crucial negative regulator of the JAK/STAT pathway, known as SOCS3 (suppressor of cytokine signaling 3). Interestingly, the exogenous upregulation of SOCS3 was capable of re-sensitizing recipient cells to gemcitabine which were already exposed to EVs derived from resistant NSCLC cells. The authors also reported the influence of miR-222-3p and EVs derived from gemcitabine resistant NSCLC cells in an in vivo model of lung metastasis. The results demonstrated that the injection of resistant NSCLC cells along with EVs derived from these cells could efficiently promote tumor metastases to other organs. Most importantly, resistant NSCLC cells that exogenously displayed low levels of miR-222-3p diminished the development of metastasis in approximately 50% of animals that were afterwards inoculated with EVs derived from gemcitabine-resistant NSCLC cells. Subsequently, the authors sought to evaluate the relevance of exosomal miR-222-3p in NSCLC patients that had undergone a gemcitabine-platinum based treatment without surgery (*n* = 50). Results from this prospective study demonstrated that exosomal miR-222-3p in human plasma correlated with a negative response to gemcitabine and disease progression in patients with advanced NSCLC. Therefore, these results suggested that miR-222-3p may possibly predict a deficient response towards this type of chemotherapeutic drug in patients with NSCLC [43].

Another study that also looked at NSCLC explored the role of exosomal miR-425-3p in a retrospective study involving a small cohort of patients (*n* = 19) [1]. Ma and collaborators compared the presence of this exo-miRNA in serum samples obtained from the same NSCLC patients at the beginning of the treatment (non-resistant), and at the date in which disease progression was diagnosed (resistant). The authors also evaluated the same serum samples originating from patients that received a first cycle of cisplatin treatment and compared them to paired serum samples obtained from patients after the last cycle of treatment, before being diagnosed as drug-resistant (*n* = 15). Results demonstrated that the level of exo-miR-425-3p was significantly higher in patients with progressive disease compared to baseline or pre-therapy status. It also had abundant expression in circulating exosomes derived from patients which had undergone the last cycle of chemotherapy when compared to their first cycle of treatment [1]. These results led the authors to further explore whether cisplatin treatment could induce the secretion of NSCLC exosomes enriched in miR-425-3p in vitro. Indeed, short-term exposure to cisplatin promoted an increase in miR-425-3p levels in NSCLC cells and induced the secretion of EVs derived from these cisplatin-treated NSCLC cells in a concentration-dependent manner. In this regard, authors also revealed that a high absolute concentration of miR-425-3p was found in exosomes derived from NSCLC cells treated with cisplatin compared to the levels of this miRNA in exosomes derived from untreated NSCLC cells. These results were also observed in three NSCLC cell lines that had undergone a long-term exposure to cisplatin and become resistant to this chemotherapeutic drug versus parental NSCLC cells. Thus, this suggests that miR-425-3p expression is promoted by cisplatin and may be linked with the development of chemoresistance. Additional experiments demonstrated that recipient NSCLC cells incubated with exosomes derived from NSCLC cells that were already treated with cisplatin became less sensitive towards treatment with this chemotherapeutic drug. Particularly, the use of a miR-425-3p inhibitor was capable of re-establishing sensitivity towards cisplatin in recipient cells previously incubated with EVs derived from cisplatin-treated NSCLC cells. Mechanistically, the authors established that cisplatin treatment may lead to increased expression of c-Myc via Wnt/β-catenin signaling in NSCLC cells. Therefore, cMyc, known as a trans activator, would bind and promote the expression of cellular and exosomal miR-425-3p, and exosomal miR-425-3p would enable autophagy via AKT1 targeting in recipient cells [23], thus enabling the initial steps towards the development of chemotherapy resistance [1].

Notably, a retrospective study [1] which analyzed the role of miR-425-3p was initially found by the authors amongst a panel of 6 exosomal miRNAs that were differentially expressed in the serum of platinum-resistant NSCLC patients (*n*= 10) when compared to serum samples obtained from platinum-sensitive NSCLC patients (*n* = 10), by HiSeq deep-sequencing analyses [23]. Interestingly, exosomal miR-425-3p was found to be significantly upregulated in the serum of platinum-resistant NSCLC patients (*n*= 76) when compared to serum samples obtained from platinum-sensitive NSCLC patients (*n*= 94). Upregulation of exo-miR-425-3p in the serum of advanced NSCLC patients was correlated with shorter progression-free survival of these patients (*p* < 0.0001). Therefore, these results and those presented months later by the same group of collaborators highlight the potential role of exo-miR-425-3p as an effective predictive biomarker for low responsiveness towards platinum-based therapy in NSCLC patients [1,23]. 

An earlier report identified another exosomal miRNA found in the serum of patients with NSCLC who had undergone a platinum-based therapy [15]. Low expression of exosomal miR-146a-5p was associated with shorter progression-free survival (*p* = 0.0014), which suggested that this exosome-derived miRNA may also aid in predicting the efficacy of cisplatin treatment in NSCLC patients [15]. However, the role of exosomal miR-146a-5p is still under debate, since other results by the same authors indicated that no effect was observed in cisplatin response when miR-146a-5p was either silenced or overexpressed in NSCLC cells [23]. 

#### 3.2.6. Multiple Myeloma

Zhang et al. examined the role of a specific set of exosomal miRNAs (miR-16-5p, miR-15a-5p, miR-20a-5p and miR-17-5p), derived from the serum of patients with multiple myeloma (MM) that had developed resistance against the use of novel therapeutic drugs, such as Bortezomib (Bz) (Table 3) [46]. Most patients diagnosed with MM will eventually experience relapse or have refractory MM due to the acquisition of drug resistance. Therefore, Zhang and colleagues attempted to assess the impact of miRNA-containing exosomes in favoring, and possibly predicting the development of drug resistance in MM patients. Authors initially designed their study using a real-world study method, which involved the use of non-randomized clinical data from a cohort of 204 MM patients treated with novel chemotherapeutic drugs, such as Bz (*n* = 115), Thalidomide (*n* = 67), or Lenalidomide (*n* = 22). Initial results revealed that Bz treatment displayed the lowest drug resistance frequency (36.5%) compared to the use of Thalidomide or Lenalidomide in these patients. In addition, this data was also accompanied by the fact that Bz is the most widely used drug to treat MM [46]. Therefore, the authors continued exploring the miRNA contents of serum-derived exosomes from peripheral blood samples obtained from patients with MM treated with Bz. Detailed information regarding these patients led to further separating the available data into Bz-responsive patients (*n* = 73) and Bz-resistant patients (*n* = 42). Subsequently, profiling data analysis predicted the presence of 83 upregulated miRNAs and 88 down-regulated miRNAs in Bz-resistant patients when compared to Bz-responsive patients, among a total of 3180 miRNAs detected on microarray (Exiqon). Only ten exosomal miRNAs were further selected according to the literature, and to their high or low abundance in the samples. Amongst the top ten miRNAs, authors found that miR-513a-5p, miR-20b-3p, and let-7d- 3p were upregulated and miR-125b-5p, miR-19a-3p, miR-21-5p, miR-20a-5p, miR-17-5p, miR-15a-5p, and miR-16-5p were found to be downregulated. Most importantly, results also revealed that a subset of the down-regulated miRNA candidates, miR-16-5p, miR-15a-5p, miR-20a-5p, and miR-17-5p, exhibited a synergistic effect in the mechanisms associated with resistance to Bz in MM patients, when a miRNA-RNA regulatory signaling network was constructed by Gene Ontology and integrated network analyses. However, authors discussed that additional studies, and a much larger cohort of MM patients experiencing drug resistance versus drug-responsive patients, is needed to confirm the importance of the downregulation of this selected set of miRNAs involved in promoting resistance to Bz. Authors agree on that this study certainly gives insight into an in-depth understanding of the in vivo intercellular crosstalk occurring in MM patients [46].

#### 3.2.7. Lymphoma

Three prospective studies were included in this section which evaluated several exosomal miRNAs initially isolated from plasma [48] or serum [24,44] using precipitating reagents, such as Exospin^TM^ [48] or ExoQuick [24,44]. Additionally, all three studies focused on predicting role of these exosomal miRNAs in chemoresistance in Diffuse large B-cell lymphoma (DLBCL) patients that were currently undergoing or had undergone a R-CHOP regimen (cyclophosphamide, doxorubicin, vincristine, and prednisone, combined with rituximab), which is the standard treatment for this type of cancer. 

One of these studies included next generation sequencing (NGS) data obtained from the group of Feng, who first reported that among 678 identified exosomal miRNAs, 37 exosomal miRNAs were found to be upregulated and 17 exosomal miRNAs were significantly downregulated in R-CHOP-resistant DLBCL cells when compared to parental DLBCL cells. Subsequently, authors sought to further gather and validate four upregulated exosomal miRNAs (miR-99a-5p, miR-125b-5p, miR-10a-5p, and miR-10b-5p) via qRT-PCR (Table 3). Results revealed that only miR-99a-5p and miR-125b-5p were significantly expressed (*p* < 0.05) in vitro. This led to the authors analysing the presence of exosomal miR-99a-5p and miR-125b-5p in serum samples obtained from a cohort of 116 patients with DLBCL treated with a R-CHOP regimen, which were later divided according to their response to treatment. Notably, results demonstrated that these serum-derived exo-miRNAs were observed to be significantly upregulated in chemoresistant DLBCL patients with progressive or stable disease (*n* = 33) when compared to chemosensitive (*n* = 83) DLBCL patients with complete or partial response to therapy (*p* < 0.001). This suggested that elevated levels of exosomal miR-99a-5p and miR-125b-5p can be used as predictors of chemotherapeutic efficacy in patients with DLBCL. In addition, high expression of these miRNAs also contributed to unfavorable prognostic factors for DLBCL patients (*p* < 0.001). Authors also emphasized that these miRNAs had an enhanced predictive capability when used in combination with the International Prognostic Index (IPI; also known as a clinical tool which is used to predict the prognosis of patients with aggressive non-Hodgkin’s lymphoma) in comparison to either one of these markers being used alone (combined predictive value of miR-99a-5p and IPI score displayed a AUC value of 0.8326; *p* < 0.001, and the combined predictive value of miR-125b-5p and IPI score displayed a AUC value of 0.8143; *p* < 0.001) [24]. 

Additionally, Xiao and collaborators evaluated the role of exosomal miR-451a in serum samples obtained from a cohort of 89 patients with DLBCL before and after undergoing a R-CHOP regimen [44]. Patients that underwent treatment were afterwards divided into complete response (*n* = 49), partial response (*n* = 24), and non-response (*n* = 25) groups. Interestingly, patients that had already undergone a R-CHOP regimen displayed elevated levels of exo-miR451a when compared to patients before treatment; however, these levels were still much lower than the levels of exo-miR451a observed in healthy subjects (*n* = 48) (AUC for ROC of 0.8038). Altogether, these results led the authors to suggest that exosomal miR-451a may be an ideal candidate, which would be capable of predicting the efficacy of using a R-CHOP regimen in DLBCL patients [44].

Zare and collaborators demonstrated that one of the most frequently overexpressed miRNAs in DLBCL [47], known as miR-155, may possibly predict R-CHOP regimen ineffectiveness by detecting their presence in exosomes, since this exo-miRNA was found to be increased in patients with DLBCL that experienced cancer relapse or were refractory to R-CHOP treatment (*n* = 16), compared to responsive DLBCL patients (*n* = 17), or DLBCL patients undergoing an R-CHOP regimen which later became responsive patients (*n* = 15) [48]. Nonetheless, the authors also agree that to establish exosomal miR-155 as a predictor of chemotherapy effectiveness, additional studies are needed that include the use of a much larger cohort of refractory/relapsed and responsive DLBCL patients to R-CHOP treatment.

## 4. Discussion

The role of exosomes as extracellular carriers of miRNAs associated with chemotherapy response is the focus of this systematic review, and recent literature has reported the expression of several miRNA candidates. Specifically, miR-21, miR-222, and miR-155, were noted to be associated with chemoresistance in lymphoma, colorectal, ovarian, breast, and lung cancer.

Interestingly, miR-21 and its two mature products, miR-21-5p and miR-21-3p, were found to be upregulated in six studies in this systematic review (Table 2). Specifically, exosomal miR-21 derived from serum was associated with low overall survival and found to be highly expressed in advanced colorectal cancer (CRC) patients with chemoresistance [26,27]. Furthermore, miR-21 was also elevated in the plasma of ovarian cancer patients that were resistant to platinum-based therapies such as carboplatin [5] or cisplatin [34], and anti-microtubule agents such as paclitaxel [5,35]. 

Intriguingly, in vitro and in vivo experiments have also demonstrated that exosomal miR-21 is capable of decreasing paclitaxel sensitivity in recipient cells or organisms with ovarian cancer tumors sensitive to chemotherapy [35]. Furthermore, miR-21 expression was greater in exosomes obtained from breast cancer patients, and elevated expression was correlated with increasing tumor size in patients undergoing chemotherapy [38]. 

Additionally, several studies also indicate that miR-21 can modulate the chemosensitivity of cancer cells by targeting tumor-suppressors such as PTEN [60], programmed Cell Death 4 (PDCD4) [61,62], and FasL via serum miR-21 [63]. Recent data has also revealed that exosomal miR-21 can regulate the TETs/PTENp1/PTEN pathway to promote cell growth in hepatocellular carcinoma [64]. In addition, the oncogenic role of this exosomal miRNA has also been found to promote chemoresistance via targeting the tumor suppressor PDCD4 in colon adenocarcinoma cells [65].

Another miRNA that was repeatedly described in this systematic review (Table 2 and Table 3) was miR-222, which was also found to be correlated with poor overall survival in advanced colorectal cancer patients resistant to oxaliplatin, leucovorin, and 5-fluorouracil [26]. A similar pattern of expression was observed for the plasma-derived exosomal miR-222 in NSCLC cancer patients, in which the upregulation of this miRNA was associated with a poor response towards gemcitabine and correlated with progressive disease in these patients [43]. Rodriguez-Martinez and collaborators evaluated the role of the serum-derived exosomal miR-222 in chemotherapy response in breast cancer patients by examining breast cancer patient samples prior to neoadjuvant therapy, which included the use of several chemotherapeutic drugs, such as doxorubicin/cyclophosphamide, docetaxel, trastuzumab, tamoxifen, anastrozole, letrozole, and/or goserelin. 

Exosomal miR-222 was found to be associated with clinical and pathological variables, such as the proliferation marker Ki67 and indicated the presence of circulating tumor cells in breast cancer patients [38]. Additionally, exosomal miR-222 is capable of conferring drug resistance to carboplatin [66], cisplatin, or docetaxel in breast cancer cells [67], and may promote adriamycin resistance in these cells via the PTEN/Akt/p27kip1 pathway [68]. Thus, this may indicate that exosomal miR-222 predicts a deficient treatment response in breast cancer patients prior to neoadjuvant chemotherapy and subsequent tumor resection [38]. 

Exosomal miR-155 was identified as having different roles in several cancers (Table 2). In DLBCL patients [48], an upregulation of this exo-miRNA was found in relapsed or non-responsive patients to R-CHOP regimen [48], and displayed a significant correlation with diminished disease-free survival in patients treated with gemcitabine [41]. Furthermore, Carvalho et al. demonstrated that resistant breast cancer and cancer stem cells were capable of transferring exosomal miR-155 to recipient cells and promoting chemoresistant traits in these cells [69]. On the other hand, Stevic and colleagues revealed that an elevated expression of miR-155 could predict an efficient response towards neoadjuvant therapy using paclitaxel, doxorubicin, and carboplatin in breast cancer patients [37].

Interestingly, there were a relatively low number of EV miRNAs identified in at least 21 research articles (Table 2) that were included in this review [1,15,17,22,24,25,26,27,28,29,30,31,32,33,34,35,36,40,41,42,43,44,45,48]. One of the potential reasons could be due to the studies proceeding to evaluate selected EV miRNA candidates that are established in the literature [1,15,17,22,25,26,27,28,31,32,36,38,41,43,44,45,48]. Another reason may be the use of a chip or card for microarray analysis that narrows the possibilities of finding an abundant set of EV miRNA candidates associated with chemoresistance in these studies [37,39,42,46]. 

An additional feature that was found among the studies that were included in this review was that different EV miRNAs were identified and described when evaluating the same type of cancer (Table 2 and Table 3). Potential reasons that could account for the discrepancy would be that these studies used different EV isolation protocols, leading to the obtainment of different EV subpopulations [70], and, thus, different EV miRNAs in the same type of cancer. The latter was evidently seen in the case of Kuhlmann and collaborators [5] who used ExoQuick to isolate EVs, compared to Yeung and collaborators [35] who used an ultracentrifugation protocol to purify these vesicles. Both studies focused on ovarian cancer, yet obtained different miRNAs. Other reasons may include the type of RNA extraction protocol that was used, or even the type of starting biofluid that was chosen to evaluate the presence of candidate miRNAs in EVs [71,72]. Thus, this would mean that a different set of EV miRNAs may have been obtained, depending on the starting biofluid or RNA extraction method used, even when the same cancer type was evaluated and the same EV isolation technique was being employed. This was observed in the case of Zhu and collaborators [36], who evaluated the presence of an EV miRNA candidate in the serum of ovarian cancer patients, and this miRNA was not found by Kuhlmann and collaborators [5], who analyzed the presence of EV miRNA candidates in the plasma of ovarian cancer patients. The aforementioned features may also aid in explaining why different cancer studies included in this review did not identify the same set of EV-associated miRNAs. One of the major reasons underlying these discrepancies may owe to the fact that miRNAs are tissue-specific and may potentially be used as diagnostic biomarkers to identify different subtypes of cancers. Furthermore, miRNAs may even be specific to the type of response to chemotherapy, including chemosensitivity, or development of chemoresistance [73,74]. 

## 5. Conclusions

This systematic review identified three key exosomal miRNAs, miR-21, miR-222, and miR-155, as being associated with chemoresistance. Specifically, these miRNAs were noted in colorectal cancer, ovarian cancer, breast cancer, and DLBCL. Whilst miR-21 and miR-222 were found to be involved in facilitating chemoresistance, miR-155 had contrasting roles, depending on the primary cancer. Elevated levels of miR-155 correlated with chemoresistance in DLBCL, whereas it was found to be predictive of a promising response to chemotherapy, in breast cancer. Therefore, further research focusing on the roles of these miRNAs in aiding and/or inhibiting chemoresistance, in colorectal-, ovarian-, and breast cancer, would be beneficial. 

## Figures and Tables

**Figure 1 cancers-13-04608-f001:**
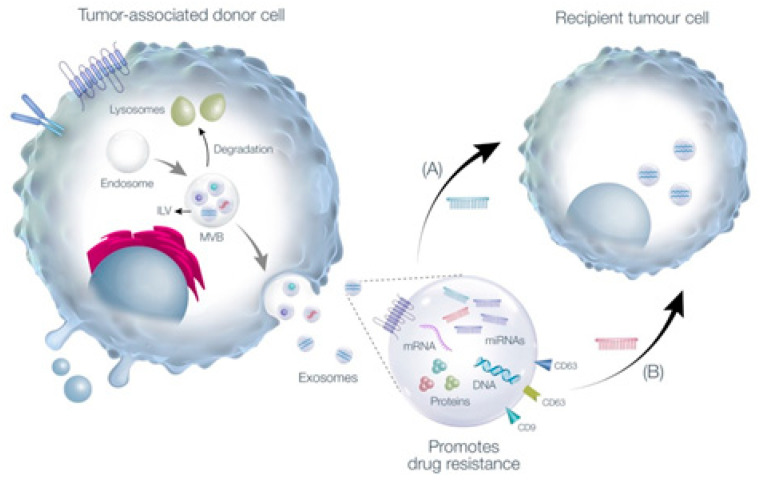
Tumor-associated cells (cancer stem cells, tumor specific cells, tumor-associated fibroblasts, or cancer- associated macrophages) have been described in the literature as being capable of promoting chemoresistance in surrounding and distant recipient cells [18]. Furthermore, resistance to chemotherapy can be promoted through the transfer of miRNAs, which are specifically sorted into intraluminal vesicles (ILV), present in multivesicular bodies (MVB) within the donor cell. Subsequently, exosome release takes place when these MVBs fuse with the plasma membrane, instead of continuing their path towards the lysosome for degradation. Most importantly, the downregulation or upregulation of distinct exosomal-derived miRNAs, as indicated in blue (A) and red (B), has also been described as vital in establishing a fine balance, which may be capable of favoring drug resistance by promoting the development of several drug resistance mechanisms, such as drug efflux, DNA damage repair, altered drug metabolism, energy programming deregulation, and epigenetic changes in recipient cells present in the tumor microenvironment [19].

**Figure 2 cancers-13-04608-f002:**
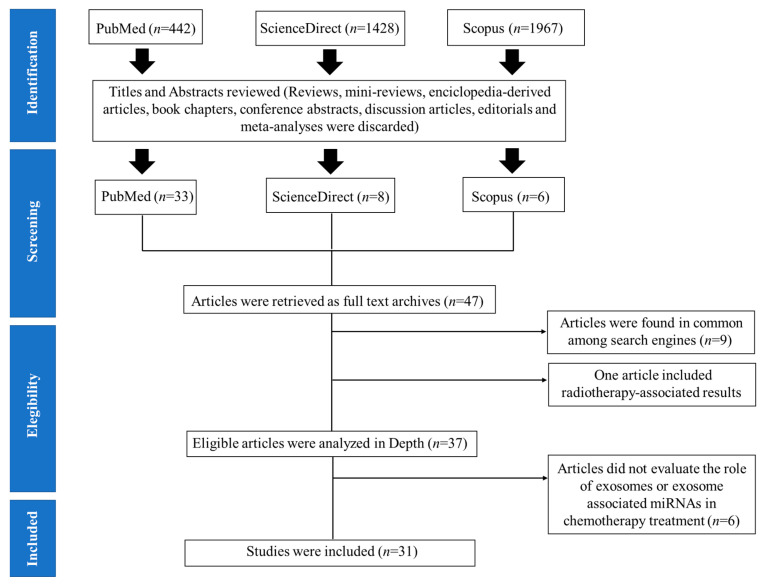
Overview of the literature study and selection process.

**Table 1 cancers-13-04608-t001:** Materials and Methods.

Publication	Search Criteria
Language	English
Time period	January 2010–September 2020
Subject	Human
Study type	Retrospective, Cohort, and Case-control
Excluded	Reviews, Encyclopedia-derived articles, Book chapters, Conference abstracts, Discussion articles, Editorials, Mini-reviews, and Meta-analyses
Keywords	Extracellular vesicles, Exosomes, EVs, miRNA, Chemotherapy, Chemoresistance, Cancer recurrence

**Table 2 cancers-13-04608-t002:** List of miRNAs involved in chemoresistance or chemosensitivity that were found amongst the set of research articles included in this review. miRNAs that were found in multiple studies of the same type of cancer have been bolded. Abbreviations: **UC**, Ultracentrifugation; **FU** or **5-FU**, 5-Fluorouracil; **DLBCL**, Diffuse large B-cell lymphoma; **HCC**, Hepatocellular carcinoma; **OXA**, Oxaliplatin; **GEM**, Gemcitabine; **TCGA**, The Cancer Genome Atlas; **NSCLC**:, Non-small cell lung cancer; **GEO**, Gene Expression Omnibus; **CMM**, Cutaneous Malignant Melanoma; **MAPKis**, Mitogen activated-protein kinase pathway inhibitors; **GBM**, Glioblastoma multiforme; **TMZ**, Temozolomide; **CSF**, cerebrospinal fluid; **BZ**, Bortezomib.

Research Article	Type of Cancer	Drug	Biofluid	Exosomal miRNA	EV Isolation	miRNA Profiling	Findings
**Bovy et al., 2015** [40]	Breast	Cyclophosphamide or Fluouracil; Epirubicin; Docetaxel or Paclitaxel	Plasma	Plasmatic miR-503	None	qRT-PCR	↑ Only after neoadjuvant therapy.
**Rodriguez-Martínez et al., 2019** [38]	Breast	Doxorubicin/cyclophosphamide Docetaxel, Trastuzumab, Tamoxifen, Anastrozole, Letrozole, Goserelin	Serum	miR-21, miR-105, miR-221 and miR-222	UC	qRT-PCR	↑ miR-21 and miR-105 in cancer patients versus healthy donors. Exosomal miRNA-222 levels correlated with clinical and pathological variables such as progesterone receptor status (*p* = 0.017) and Ki67 (*p* = 0.05). ↓ miR-221 during neoadjuvant therapy in patients with compromised lymph nodes.
**Stevic et al., 2018** [37]	Breast	Paclitaxel, Doxorubicin and Carboplatin	Plasma	miR-155, miR-301, miR-27a, miR-376a miR-376c	ExoQuick	Microarray	↑ miR-155 and miR-301 predicted efficient response towards neoadjuvant therapy.
**Zhong et al., 2016** [39]	Breast	Docetaxel, Epirubicin, Pemetrexed disodium and, Cytoxan	Tumor tissue	miR-574, miR-210-3p, miR-138-5p, miR-4258, miR-744-5p, miR-7107-5p, miR-6780b-3p, miR-3178, miR-4298, miR-423-5p, miR-7847-3p and miR-4443	None	qRT-PCR	↑ miR-574-3p associated with progressive disease.
**De Miguel Pérez et al., 2020** [26]	Metastatic colorectal cancer	FOLFOX-6m (OXA, Leucovorin, 5-FU) plus Bevacizumab	Serum	**miR-21** miR-92a miR-222 miR-19b	UC	qRT-PCR	↑ marker for diagnosis and associated with low overall survival.
**Jin et al., 2019** [27]	Advanced colorectal cancer	OXA, 5-FU, (and leucovorin)	Serum	**miR-21-5p**, miR-1246, miR-1229-5p and miR-96-5p	UC	qRT-PCR	↑ chemoresistant patients versus chemosensitive patients.
**Liu et al., 2019** [25]	Colorectal	OXA	Tumor tissue	miR-128-3p	None	qRT-PCR	↓ relapse after therapy versus patients which responded well to neoadjuvant therapy.
**Feng et al., 2019** [24]	DLBCL	R-CHOP regimen (cyclophosphamide, doxorubicin, vincristine, and prednisone, combined with the anti-CD20 monoclonal antibody–rituximab).	Serum	miR-99a-5p miR-125b-5p	ExoQuick	qRT-PCR	↑ chemoresistant group vs chemosensitive group.
**Xiao et al., 2019** [44]	DLBCL	Rituximab combined with Chemotherapy (R-CHOP regimen)	Serum	miR-451a	ExoQuick	qRT-PCR	↓ miR-451a in cancer patients versus healthy subjects and may predict treatment efficacy.
**Zare et al., 2019** [48]	DLBCL	R-CHOP regimen	Plasma	miR-155	ExoSpin	qRT-PCR	↑ miR-155 in relapsed patients or non-responsive to R-CHOP regimen.
**Hu et al., 2019** [29]	Gastric	Non-specified	Ascites	miR-760, miR-6821-5p, miR-4745-5p, miR-200a-5p, miR-4741 and miR-320	UC	RNA sequencing	↑ Progressive disease
**Yang et al., 2017** [28]	GBM	TMZ (only evaluated in vitro)	Serum and tumor tissue	miR-221	ExoQuick	qRT-PCR	↑ miR-221 may predict TMZ resistance.
**Zeng et al., 2018** [45]	GBM	TMZ	Serum and CSF	miR-151a	UC	qRT-PCR	↓ miR-151a prior to therapy was associated with poor TMZ response and poor prognosis (CSF fluid).
**Fu et al., 2018** [31]	HCC	5-FU, OXA, GEM, and Sorafenib	Tumor tissue	miR-32-5p	None	qRT-PCR	↑ tumor tissue and associated with short overall survival and progression-free survival.
**Wang et al., 2019** [17]	HCC	Sorafenib (in vitro use only)	Serum and tumor tissue	miR-744	UC	qRT-PCR	↓ miR-744 in HCC tissues and exosomes from serum of these patients.
**Svedman et al., 2018** [42]	Metastatic BRAFV600 mutated CMM	MAPKis	Plasma	let-7g-5p and miR-497-5p	MiRCURY Exosome Isolation Kit	Microarray (MiRCURY)	↑ let-7g-5p during treatment associated with improved disease control. ↑ miR-497-5p during treatment associated with prolonged progression-free survival.
**Zhang et al., 2016** [46]	Multiple myeloma	Bz, Thalidomide and lenalidomide	Serum	miR-16-5p, miR-15a-5p, miR-20a-5p, and miR-17-5p	UC	Microarray (MiRCURY)	↓ miR-16-5p, miR-15a-5p, miR-20a-5p, and miR-17-5p in patients resistant to Bz.
**Ma et al., 2019** [1]	NSCLC	Cisplatin	Serum	**miR-425-3p**	ExoQuick	qRT-PCR	↑ platinum-resistant cancer patients.
**Wei et al., 2017** [43]	NSCLC	GEM	Plasma	miR-222-3p	UC	qRT-PCR	↑ miR-222-3p associated with low response towards chemotherapy and progressive disease.
**Yuwen et al., 2019** [23]	NSCLC	Cisplatin	Serum	**miR-425-3p**, miR-1273h, miR-4755-5p, miR-9-5p, **miR-146a-5p**, and miR-215-5p	ExoQuick	RNA sequencing and qRT-PCR	↑ miR-425-3p associated with low response and poor progression-free survival.
**Yuwen et al., 2017** [15]	NSCLC	Cisplatin	Serum	**miR-146a-5p**	ExoQuick	RNA sequencing and qRT-PCR	↓ miR-146a-5p associated with shorter progression-free survival.
**Alharbi et al., 2020** [33]	Ovarian	Not Indicated	Plasma	miR-891-5p	UC	qRT-PCR	↑ relapse
**Kanlikilicer et al., 2018** [32]	Ovarian	Paclitaxel (in vitro and in vivo use only)	Tumor tissue and TCGA tissue samples	miR-1246	None	qRT-PCR	↑ Associated with worse overall prognosis.
**Kuhlmann et al., 2019** [5]	Ovarian	Carboplatin and Paclitaxel	Plasma	miR-181a, miR-1908, **miR-21**, miR-486 and **miR-223**	ExoQuick	RNA sequencing	↑ platinum-resistant cancer patients (preliminary study).
**Pink et al., 2015** [34]	Ovarian	Cisplatin (in vitro use only)	Tumor tissue	**miR-21-3p**	None	GEO repository data and TCGA data	↑ tumor tissue associated with shorter progression-free interval.
**Yeung et al., 2016** [35]	Ovarian	Paclitaxel (in vitro and in vivo use only)	Primary culture derived from ovarian tissue	**miR-21**	UC	Next generation sequencing	miR-21 can ↓ paclitaxel sensitivity in vitro and in vivo.
**Zhu et al., 2019** [36]	Ovarian	Taxol and Cisplatin	Serum and tumor tissue	**miR-223**	ExoQuick	qRT-PCR	↓ miR-223 associated with longer progression-free survival and decreased cancer relapse.
**Mikamori et al., 2017** [41]	Pancreatic ductal adenocarcinoma	GEM	Plasma	miR-155	ExoQuick	qRT-PCR	↑ associated with low disease-free survival.
**Corcoran et al., 2014** [30]	Prostate	Docetaxel	Tumor and urine	miR-34a	None	Gene expression dataset (Omnibus)	↓ cancer tissue vs benign and recurrent cancer vs no recurrence.
**Huang et al., 2015** [14]	Prostate	Androgen-deprivation therapy (Docetaxel, abiraterone acetate, prednisone, cabazitaxel, and/or mitoxantrone.	Plasma	miR-1290, miR-1246, and miR-375	ExoQuick	RNA Sequencing	↑ Associated with poor overall survival.
**He et al., 2020** [22]	Renal cell carcinoma	Sorafenib	Plasma	miR-31-5p	UC and sucrose cushion	qRT-PCR	↑ Progressive disease during chemotherapy.

**Table 3 cancers-13-04608-t003:** List of miRNAs identified in the studies that were included in this systematic review. miRNAs found to be in common among several studies regarding the same type of cancer have been bolded.

Tumor Type	Exosomal miRNAs	References
**Colorectal**	miR-19b, **miR-21**, miR-222 and miR-92a	[25,26,27]
**Ovarian**	miR-891-5p, miR-1246, miR-181a, miR-1908, **miR-21**, miR-486, **miR-223**, miR-21-3p	[5,32,33,34,35,36]
**Breast**	miR-155, miR-301, mi-27a, miR-376a, miR-376c, miR-21, miR-105, miR-221, miR-222, miR-574-3p, miR-503	[37,38,39,40]
**Renal cell carcinoma**	miR-31-5p	[22]
**Lung**	miR-222-3p, **miR-425-3p**, **miR-146a-5p**	[1,15,23,43]
**Multiple myeloma**	miR-16-5p, miR-15a-5p, miR-20a-5p and miR-17-5p	[46]
**Lymphoma**	miR-99a-5p, miR-125b-5p, miR-451a, miR-155	[24,44,48]
**Prostate**	miR-34a, miR-1290, miR-1246, miR-375	[14,30]
**Hepatocellular carcinoma**	miR-32-5p, miR-744	[17,31]
**Gastric**	miR-760, miR-6821-5p, mi-4745-5p, miR-200a-5p, miR-4741, miR-320	[29]
**Pancreatic ductal adenocarcinoma**	miR-155	[41]
**Metastatic BRAFV600 mutated cutaneous malignant melanoma**	Let-7g-5p, miR-497-5p	[42]
**Glioblastoma**	miR-221, miR-151a	[28,45]

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
