# Peer review of "Extracellular Vesicle-Associated miRNAs and Chemoresistance: A Systematic Review"

_cancers, 2021, doi:10.3390/cancers13184608_

Round 1
Reviewer 1 Report
The review by Campos et al. offers a methodologically clear and precise perspective on the current evidence on the association between exosomal miRNA and chemotherapy. The potential of exosomal miRNA as biomarkers, including their potential as chemotherapy resistance indicators, is huge, but there are many studies with insufficient methodological details, so this kind of accurate revision may be useful in the field. However, some parts or sentences are confused and needs to be introduced in cleared manner, otherwise the work would be not sufficiently useful. Besides, some critical issues underlying the difficulties in identifying miRNAs as biomarkers, as for example the reproducibility problems among different studies, are not discussed.
Major points
In introducing miRNAs, authors first explain that “They can be released into the blood stream in response to several physiological processes…”, then they explain that “…these molecules are capable of binding to complementary sequences in the 3`untranslated region of their target mRNAs, thus leading to mRNA inhibition”. The order of the sentence is misleading, as the first function of miRNAs is not being released into blood stream, as they all have the intracellular function to modulates mRNA translation. The order of the two sentences should be inverted, to offer readers a proper introduction first to intracellular function and then to their extracellular release. In the context, the distinction between “free” circulating miRNA and miRNA encapsulated in extracellular vesicles should be also clearer.
Please specify the biophysical properties that allows to classify EVs “into exosomes, microvesicles, or apoptotic bodies”, as the line 77 sentence is confusing.
The Line 128 sentence about the exclusion criteria “studies were removed based on the following exclusion criteria, which included studies focusing only on in vitro and in vivo results,” is unclear, what do authors mean for “only…in vivo results”? Studies on animals? Association studies on patients?
Line 159. The sentence “Recent literature suggests that EV- or exo-miRNAs may predict the emergence of 159 chemotherapy resistance [1, 14, 21, 22].” Follows the illustration of the selection for relevant papers, and it is unclear whether references 1, 14, 21, 22 are part of these selected papers or not. Authors should better introduce the content of the papers that passed the selection: type of cancers, type of study etc. from lines 165 to 172 only a few references included, and even in this case it is not clear if these papers were included in the selection.
The selection criteria of Table 3 is unclear. What do authors mean for “evaluated”? The most interesting? Those more relevant in terms of different expression? Those identified in more than one study?
Authors dedicate the Discussion to miRNAs identified in more than one study (and more than one cancer type). However, they did not comment on the reasons underlying this relatively low number of miRNAs identified in more than one study or on the evidence that in the same cancers very often different miRNAs are identified. There could be technical problems in amplifying miRNAs or in isolating exosomes and/or circulating miRNAs?
Minor points
Please check the sentence “Summarized graphical abstract indicating that a distinct signature of exosome-derived miRNAs is involved in the development of chemotherapy resistance in each type of cancer evaluated in this systematic review.”, as it is unclear/incorrect. Overall, the simple summary is difficult to catch
Introduction. Please add a specific reference at the end of the sentence “Additionally, resistance has also been shown to be reversible, and it is highly likely that epigenetic factors are involved in facilitating resistance.”
Line 177 Eliminate comma from “drugs such as, Oxaliplatin,…”
Line 180-183 The description of the study by Liu is unclear, why adding this description if there is a detailes description below (lines 205-219)?
In the Discussion, authors wrote that “Specifically, miR-21, miR-222, and miR-155, 616 were noted to be associated with chemoresistance in multiple cancers.” Could authors recapitulate what cancers, in the context of the Discussion?
Author Response
We thank the reviewer for their insightful comments and questions relating to the manuscript. We have carefully considered their comments and addressed them in detail. Please see below.
Reviewer 1
The review by Campos et al. offers a methodologically clear and precise perspective on the current evidence on the association between exosomal miRNA and chemotherapy. The potential of exosomal miRNA as biomarkers, including their potential as chemotherapy resistance indicators, is huge, but there are many studies with insufficient methodological details, so this kind of accurate revision may be useful in the field. However, some parts or sentences are confused and needs to be introduced in cleared manner, otherwise the work would be not sufficiently useful. Besides, some critical issues underlying the difficulties in identifying miRNAs as biomarkers, as for example the reproducibility problems among different studies, are not discussed.
Major points
In introducing miRNAs, authors first explain that “They can be released into the blood stream in response to several physiological processes…”, then they explain that “…these molecules are capable of binding to complementary sequences in the 3`untranslated region of their target mRNAs, thus leading to mRNA inhibition”. The order of the sentence is misleading, as the first function of miRNAs is not being released into blood stream, as they all have the intracellular function to modulates mRNA translation. The order of the two sentences should be inverted, to offer readers a proper introduction first to intracellular function and then to their extracellular release. In the context, the distinction between “free” circulating miRNA and miRNA encapsulated in extracellular vesicles should be also clearer.
Reply: The order of the two sentences has been changed, as requested. In addition, the distinction between “free” circulating miRNA and miRNA encapsulated in extracellular vesicles has also been written in a clearer manner.
Please specify the biophysical properties that allows to classify EVs “into exosomes, microvesicles, or apoptotic bodies”, as the line 77 sentence is confusing.
Reply: Line 77 has been modified in order to specify which biophysical properties can be used to distinguish exosomes from microvesicles or apoptotic bodies.
The Line 128 sentence about the exclusion criteria “studies were removed based on the following exclusion criteria, which included studies focusing only on in vitro and in vivo results,” is unclear, what do authors mean for “only…in vivo results”? Studies on animals? Association studies on patients?
Reply: Line 128 has been modified to clarify that studies which solely included animal model-based results were excluded from this systematic review.
Line 159. The sentence “Recent literature suggests that EV- or exo-miRNAs may predict the emergence of 159 chemotherapy resistance [1, 14, 21, 22].” Follows the illustration of the selection for relevant papers, and it is unclear whether references 1, 14, 21, 22 are part of these selected papers or not. Authors should better introduce the content of the papers that passed the selection: type of cancers, type of study etc. from lines 165 to 172 only a few references included, and even in this case it is not clear if these papers were included in the selection.
Reply: Line 159 has been modified as suggested. The contents of the selected research articles (31 articles) were introduced, including the cancer types and study designs that were involved. References of all 31 articles were included in this paragraph.
The selection criteria of Table 3 is unclear. What do authors mean for “evaluated”? The most interesting? Those more relevant in terms of different expression? Those identified in more than one study?
Reply: Table 3 shows a specific set of exosomal miRNAs that are associated with a specific cancer type, independent of the type of chemotherapy that was being used on the set of cancer patients.
Authors dedicate the Discussion to miRNAs identified in more than one study (and more than one cancer type). However, they did not comment on the reasons underlying this relatively low number of miRNAs identified in more than one study or on the evidence that in the same cancers very often different miRNAs are identified. There could be technical problems in amplifying miRNAs or in isolating exosomes and/or circulating miRNAs?
Reply: A new section was included in the Discussion in order to explain the potential reasons behind why a low number of miRNAs were identified in these studies, and why different miRNAs were identified in studies focusing on the same type of cancer. Additional references were included.
Minor points
Please check the sentence “Summarized graphical abstract indicating that a distinct signature of exosome-derived miRNAs is involved in the development of chemotherapy resistance in each type of cancer evaluated in this systematic review.”, as it is unclear/incorrect. Overall, the simple summary is difficult to catch
Reply: Sentence has been modified in order to give a better understanding of the findings obtained in this systematic review.
Introduction. Please add a specific reference at the end of the sentence “Additionally, resistance has also been shown to be reversible, and it is highly likely that epigenetic factors are involved in facilitating resistance.”
Reply: A specific reference has been included as requested.
Line 177 Eliminate comma from “drugs such as, Oxaliplatin,…”
Reply: Comma has been deleted in this sentence as requested.
Line 180-183 The description of the study by Liu is unclear, why adding this description if there is a detailes description below (lines 205-219)?
Reply: Sentence was modified so as to clearly state that differences may arise when evaluating miRNAs isolated from serum-derived EVs [28,29] compared to the study of Liu and colleagues which focused on evaluating miRNAs directly obtained from tumor tissue.
In the Discussion, authors wrote that “Specifically, miR-21, miR-222, and miR-155, 616 were noted to be associated with chemoresistance in multiple cancers.” Could authors recapitulate what cancers, in the context of the Discussion?
Reply: Sentence was modified and now includes the types of cancers which displayed miR-21, miR-222 and miR-155 that were associated with chemoresistance.
Reviewer 2 Report
The review article Extracellular vesicle-associated miRNAs and Chemoresistance: A Systemic Review by Salomon et al. is well written and provides an in-depth analysis of miRNAs expression and value with respect to chemoresistance in various cancer types.
There are some minor mistakes in the document mostly typos, spacing and flow of the table.
Abstract line 28 change to "Cancer is a leading public health issue"
Figure 1 line 93 change to Tumor
Figure 1 line 99 the colors used for A and B are not really that visible. The green looks like blue.
Line 142 what is meant by revising the data? Do you mean revisiting the data?
Lines 172, 181, 210, 211, 237, 245, 248, 249, 279, 282, 323, 333, 366, 393, 403, 461, 629 it should be spelt as tumor and not tumour
Line 207 should it be CRC cell line and not CLC?
Line 305 define APAF1
Line 366 remove "that showed" after revealed
Line 395 change to "invasive potential"
Table 2 DLBCL should be Diffuse and not Difusse
Table 2 need some formatting. Please check. The header seems to show in between pages and not on every new page. Also, for the reference Corcoran et al remove space between "tumor tissue"
Table 2 it would be good if the same cancer type is grouped together or is listed one after the other for easier comparison.
After table 2 all the page numbers are incorrect. Please change.
Line 505 place the reference [1] after the prospective study.
Table 3 header, change to Tumor
Line 631 please define PDCD4
Line 632 andFasL add a space
Line 641 abbreviate non-small cell lung cancer - NSCLC
Author Response
We thank the reviewer for their insightful comments and questions relating to the manuscript. We have carefully considered their comments and addressed them in detail. Please see below.
Reviewer 2
The review article Extracellular vesicle-associated miRNAs and Chemoresistance: A Systemic Review by Salomon et al. is well written and provides an in-depth analysis of miRNAs expression and value with respect to chemoresistance in various cancer types.
There are some minor mistakes in the document mostly typos, spacing and flow of the table.
Abstract line 28 change to "Cancer is a leading public health issue"
Reply: Part of sentence has been modified as requested.
Figure 1 line 93 change to Tumor
Reply: Word has been modified in line 93.
Figure 1 line 99 the colors used for A and B are not really that visible. The green looks like blue.
Reply: line 99 was modified in order to clarify that the color used for A is in fact blue.
Line 142 what is meant by revising the data? Do you mean revisiting the data?
Reply: Thank you for your comment. The word “revisiting” replaced “revising” in line 142.
Lines 172, 181, 210, 211, 237, 245, 248, 249, 279, 282, 323, 333, 366, 393, 403, 461, 629 it should be spelt as tumor and not tumour
Reply: Thank you, this word has been modified throughout the text.
Line 207 should it be CRC cell line and not CLC?
Reply: Indeed, it should be “CRC” instead of “CLC”.
Line 305 define APAF1
Reply: Line 305 was modified in order to define APAF1.
Line 366 remove "that showed" after revealed
Reply: The words “that showed” were removed as requested.
Line 395 change to "invasive potential"
Reply: word “invasion” was changed to “invasive” as requested.
Table 2 DLBCL should be Diffuse and not Difusse
Reply: word “Diffuse” was modified as requested.
Table 2 need some formatting. Please check. The header seems to show in between pages and not on every new page. Also, for the reference Corcoran et al remove space between "tumor tissue"
Reply: Table 2 has been modified as requested. Space was removed between “tumor tissue”.
Table 2 it would be good if the same cancer type is grouped together or is listed one after the other for easier comparison.
Reply: Thank you for your comment. Table 2 has been modified as requested.
After table 2 all the page numbers are incorrect. Please change.
Reply: Page numbers were corrected after table 2.
Line 505 place the reference [1] after the prospective study.
Reply: Line 505 was modified as requested.
Table 3 header, change to Tumor
Reply: Word was modified as requested.
Line 631 please define PDCD4
Reply: PDCD4 was defined as requested in line 631.
Line 632 andFasL add a space
Reply: Space was included between “and” and “FasL”.
Line 641 abbreviate non-small cell lung cancer – NSCLC
Reply: “Non-small cell lung cancer” was abbreviated as requested.
Reviewer 3 Report
please see the attachement
Author Response
We thank the reviewer for their insightful comments and questions relating to the manuscript. We have carefully considered their comments and addressed them in detail. Please see below.
Reviewer 3
These authors have set out to perform an important task, a review of the available published information regarding Extracellular Vesicles (and their associated miRNAs) with cancer chemoresistance. They have produced a rather comprehensive catalog of previously published material on this topic. However, they have not performed any substantive critique of that literature.
- Their paper (and abstract) extensively describe the methods of search and identification. This can be reduced (especially in the abstract).
Reply: Thank you for your comment. Methods of search and identification were extensively described in this review as well as the abstract according to the approaches recommended by Kelley and collaborators (Kelley et al., 2018). In addition, several systematic reviews published in Cancers displayed the same extensive description as observed in this manuscript review (Klicka et al., 2021; Brown et al., 2021; Boulva et al., 2021).
- Kelley, G. A., & Kelley, K. S. (2018). Systematic reviews and cancer research: a suggested stepwise approach. BMC cancer, 18(1), 1-9.
- Klicka, K., Grzywa, T. M., Klinke, A., Mielniczuk, A., & WÅ‚odarski, P. K. (2021). The Role of miRNAs in the Regulation of Endometrial Cancer Invasiveness and Metastasis—A Systematic Review. Cancers, 13(14), 3393.
- Brown, T. J., & James, V. (2021). The Role of Extracellular Vesicles in the Development of a Cancer Stem Cell Microenvironment Niche and Potential Therapeutic Targets. Cancers, 13(10), 2435.
- Boulva, K., Apte, S., Yu, A., Tran, A., Shorr, R., Song, X., ... & Nessim, C. (2021). Contemporary Neoadjuvant Therapies for High-Risk Melanoma: A Systematic Review. Cancers, 13(8), 1905.
- They need to do more summarizing, analyzing, and critiquing the data from the papers.
Reply: Thank you for your comment. An additional paragraph involving critique and analysis was included in the Discussion section. Table 2 was modified in order to improve the display of information regarding the studies associated with a particular type of cancer.
- The only substantive sentence in the abstract reads, “Specifically, miR-21, miR-222, and miR-155 dis-43 played relevant roles in chemotherapy resistance and were found to be in common in colorectal 44 cancer, ovarian cancer, breast cancer, and diffuse large B cell lymphoma patients.” What do the authors mean by “relevant roles”? What do these miRNAs do? Do they do the same things in more than one cancer type? What are potential targets? Etc.
Reply: Thank you for your comment. The relevant roles of these miRNAs are extensively described in the Discussion section.
- The authors also state, “Thus, further research regarding the roles of these miRNAs would be beneficial in terms of designing novel tools to counteract the progression of cancer in a non-to-distant future.” What does this mean? This statement deserves an entire section of the paper.
Reply: Thank you for your comment. Indeed, this statement deserves an entire section although it may be beyond the scope of this review. It would be interesting to explore the data available on the design of novel tools to counteract the progression of cancer in our next systematic review.
Reviewer 4 Report
These authors have set out to perform an important task, a review of the available published information regarding Extracellular Vesicles (and their associated miRNAs) with cancer chemoresistance. They have produced a rather comprehensive catalog of previously published material on this topic. However, they have not performed any substantive critique of that literature.
- Their paper (and abstract) extensively describe the methods of search and identification. This can be reduced (especially in the abstract).
- They need to do more summarizing, analyzing, and critiquing the data from the papers.
- The only substantive sentence in the abstract reads, “Specifically, miR-21, miR-222, and miR-155 dis-43 played relevant roles in chemotherapy resistance and were found to be in common in colorectal 44 cancer, ovarian cancer, breast cancer, and diffuse large B cell lymphoma patients.” What do the authors mean by “relevant roles”? What do these miRNAs do? Do they do the same things in more than one cancer type? What are potential targets? Etc.
- The authors also state, “Thus, further research regarding the roles of these miRNAs would be beneficial in terms of designing novel tools to counteract the progression of cancer in a non-to-distant future.” What does this mean? This statement deserves an entire section of the paper.
Author Response

(The authors gave the same response as above.)

Round 2
Reviewer 1 Report
The review by Campos et al. has improved its weaknesses, introducing critical issues
Author Response
We thank the reviewer for their insightful comments and questions relating to the manuscript.
Reviewer 3 Report
This paper is somewhat improved from the prior versions. However, I believe that it could be improved further to make a real critical contribution to the literature:
- The authors should add some of the information from the Conclusions to the abstract: This systematic review identified three key exosomal miRNAs: miR-21, miR-222, and miR-155, as being associated with chemoresistance. Specifically, these miRNAs were noted in colorectal cancer, ovarian cancer, breast cancer, and DLBCL. Whilst miR-21 and
miR-222 were shown to be involved in facilitating chemoresistance, miR-155 had contrasting roles, depending on the primary cancer. Elevated levels of miR-155 correlated with chemoresistance in DLBCL, whereas it was shown to be predictive of a promising response to chemotherapy, in breast cancer." - Tables 2 and 3 should highlight miRNAs that were identified in multiple studies of the same cancer type.
- The authors should discuss why multiple studies of the same cancer type did not identify more of the same miRNAs.
- The authors should comment on why more studies of different cancers did not identify the same miRNAs
Author Response
We thank the reviewer for their insightful comments and questions relating to the manuscript. We have carefully considered their comments in the second round of revision and addressed them in detail in the revised version of the manuscript (with track changes). Also, please see below:
This paper is somewhat improved from the prior versions. However, I believe that it could be improved further to make a real critical contribution to the literature:
- The authors should add some of the information from the Conclusions to the abstract: This systematic review identified three key exosomal miRNAs: miR-21, miR-222, and miR-155, as being associated with chemoresistance. Specifically, these miRNAs were noted in colorectal cancer, ovarian cancer, breast cancer, and DLBCL. Whilst miR-21 and miR-222 were shown to be involved in facilitating chemoresistance, miR-155 had contrasting roles, depending on the primary cancer. Elevated levels of miR-155 correlated with chemoresistance in DLBCL, whereas it was shown to be predictive of a promising response to chemotherapy, in breast cancer."
Reply: The abstract has been modified as suggested.
- Tables 2 and 3 should highlight miRNAs that were identified in multiple studies of the same cancer type.
Reply: miRNAs that were found to be in common among several studies regarding the same type of cancer were highlighted in Tables 2 and 3 as requested.
- The authors should discuss why multiple studies of the same cancer type did not identify more of the same miRNAs.
Reply: Thank you for your comment. This part of the discussion was already included in the last version of the manuscript.
- The authors should comment on why more studies of different cancers did not identify the same miRNAs
Reply: Thank you for your comment. Additional references were included to comment on the potential reasons as to why different cancer studies did not identify the same EV-associated miRNAs.
Round 3
Reviewer 3 Report
The paper has been incrementally improved and will now make a useful addition to the literature.
I recommend one final change:
The abstract should contain a sentence stating that multiple studies of the same tumor type identified different microRNAs, and few studies identified the same ones.
Author Response
We thank the reviewer for their insightful comments and questions relating to the manuscript. We have carefully considered their final comments and addressed them in the abstract.